# Biomarkers of Aggressive Prostate Cancer at Diagnosis

**DOI:** 10.3390/ijms24032185

**Published:** 2023-01-22

**Authors:** Brock E. Boehm, Monica E. York, Gyorgy Petrovics, Indu Kohaar, Gregory T. Chesnut

**Affiliations:** 1Urology Service, Walter Reed National Military Medical Center, Bethesda, MD 20814, USA; 2School of Medicine, Uniformed Services University of Health Science, Bethesda, MD 20814, USA; 3Center for Prostate Disease Research, Department of Surgery, Uniformed Services University of the Health Sciences and the Walter Reed National Military Medical Center, Bethesda, MD 20814, USA; 4Henry Jackson Foundation for the Advancement of Military Medicine (HJF), Bethesda, MD 20817, USA

**Keywords:** prostate cancer, diagnosis, biomarkers, race

## Abstract

In the United States, prostate cancer (CaP) remains the second leading cause of cancer deaths in men. CaP is predominantly indolent at diagnosis, with a small fraction (25–30%) representing an aggressive subtype (Gleason score 7–10) that is prone to metastatic progression. This fact, coupled with the criticism surrounding the role of prostate specific antigen in prostate cancer screening, demonstrates the current need for a biomarker(s) that can identify clinically significant CaP and avoid unnecessary biopsy procedures and psychological implications of being diagnosed with low-risk prostate cancer. Although several diagnostic biomarkers are available to clinicians, very few comparative trials have been performed to assess the clinical effectiveness of these biomarkers. It is of note, however, that a majority of these clinical trials have been over-represented by men of Caucasian origin, despite the fact that African American men have a 1.7 times higher incidence and 2.1 times higher rate of mortality from prostate cancer. Biomarkers for CaP diagnosis based on the tissue of origin include urine-based gene expression assays (PCA3, Select MDx, ExoDx Prostate IntelliScore, Mi-Prostate Score, PCA3-PCGEM1 gene panel), blood-based protein biomarkers (4K, PHI), and tissue-based DNA biomarker (Confirm MDx). Another potential direction that has emerged to aid in the CaP diagnosis include multi-parametric magnetic resonance imaging (mpMRI) and bi-parametric magnetic resonance imaging (bpMRI), which in conjunction with clinically validated biomarkers may provide a better approach to predict clinically significant CaP at diagnosis. In this review, we discuss some of the adjunctive biomarker tests along with newer imaging modalities that are currently available to help clinicians decide which patients are at risk of having high-grade CaP on prostate biopsy with the emphasis on clinical utility of the tests across African American (AA) and Caucasian (CA) men.

## 1. Introduction

Prostate cancer (CaP) is the second most common non-skin malignancy among men in the United States, with current estimates of 268,490 new cases of prostate cancer and 34,500 deaths, making this the second leading cause of cancer death in men in the United States [1]. Interestingly, prostate cancer has the highest incidence among men of African ancestry, followed by European and Asian men [2]. 

CaP diagnosis is based on the gold standard procedure of transrectal ultrasound (TRUS) guided needle biopsy of the prostate. The diagnostic biopsy is informed by the combination of any of the following: elevated serum prostate-specific antigen (PSA), PSA kinetics, abnormal digital rectal exam (DRE), family history, race, or abnormal previous biopsy. Recently, screening for prostate cancer using PSA has come under considerable criticism due to several trials demonstrating that PSA screening often leads to overdiagnosis and overtreatment [3,4,5]. Evidence clearly suggests that at least 30% of all prostate cancer is low risk, with another 40% belonging in the intermediate-risk category at diagnosis [6]. In addition, the use of (mpMRI) as a screening tool to allow men to avoid unnecessary prostate biopsies while improving the diagnostic accuracy of subsequent biopsies has been validated [7]. Due to these facts, identifying a biomarker to help figure out which patient is harboring clinically significant prostate cancer is of the utmost importance. In this direction, several adjunctive biomarker tests have been developed to predict the risk of clinically significant prostate cancer and prevent unnecessary prostate biopsies [8]. This could also avoid unnecessary harm in the form of anxiety, bleeding, risk of infection requiring hospitalization, and the psychological implications of being diagnosed with prostate cancer. 

An ideal biomarker has high sensitivity and specificity, reproducibility, and easy usability with quantifiable measures; is cost effective; and provides clear results for clinicians while being easily applied to different racial groups [8]. Unfortunately, there have been very few comparative trials among these biomarkers and clinicians often feel unsure as to which one provides the most useful information. Additionally, most of the biomarker studies and clinical trials are over-represented by men of Caucasian origin. In the following studies that we discuss, a racial breakdown of their patients was often not reported. However, of those that did provide a racial demographic, African American men represented fewer than 10% of all patients evaluated, despite having a vastly higher prostate cancer incidence and mortality rate [1,2,9]. Therefore, it is important to address the clinical utility of the commercially available biomarkers in the context of race.

In this review, we discuss some of the biomarker tests that are currently available to help clinicians decide which patients are at risk of having an aggressive CaP at biopsy with the focus on clinical utility of the diagnostic tests across African American (AA) and Caucasian (CA) men. 

## 2. Urine Based Diagnostic Biomarkers

### 2.1. Prostate Cancer Antigen 3 (PCA3)

*PCA3* is a prostate specific noncoding messenger RNA (mRNA) that has been found to be overexpressed in more than 90% of all prostate tumors compared to that of benign prostatic tissue [10]. The use of quantifying *PCA3* RNA in post-DRE urine has previously been reported in several studies [10,11]. Progensa *PCA3* assay (Progensa Test Kit, Hologic, Marlborough, MA, USA) is an FDA-approved diagnostic test, intended to be used in men with an elevated serum PSA and previous negative prostate biopsy results who are 50 years of age or older [12] (Table 1, Figure 1). The assay provides clinicians with a *PCA3* score, which is derived from the ratio of *PCA3* RNA molecules to PSA RNA molecules that are detected in a patient’s urinary specimen following a DRE. This specific assay is included in the European Association of Urology (EAU) and NCCN 2020 guidelines for repeat biopsy decision making.

*PCA3* has been demonstrated to have variable sensitivity and specificity, dependent upon the cut-off score chosen (*PCA3* score of 25 or 35). As of now, there is no consensus *PCA3* cut-off score. A *PCA3* score <25 is associated with less probability of prostate cancer on subsequent repeat biopsy. A *PCA3* score of 35 was associated with a sensitivity of 58% to 82%, specificity of 58% to 76%, positive predictive value (PPV) of 67% to 69%, negative predictive value (NPV) of 87%, and an area under the curve (AUC) of 0.68 to 0.87 (95% CI for 0.87: 0.81–0.92) [13,14]. A validation study of the *PCA3* urinary assay under NCI Early Detection Research Network (EDRN) in 859 men from 11 centers showed a PPV of 80% (95% CI, 72–86%) in the initial biopsy setting and an NPV of 88% (95% CI, 81–93%) in the repeat biopsy setting [15]. On the basis of these data, the use of *PCA3* in the repeat biopsy setting would reduce the number of biopsies by almost half, and only 3% of men with a low *PCA3* score would have high-grade prostate cancer that would be missed [15].

Deras et al. were able to demonstrate that *PCA3* is independent of prostate volume, serum PSA level, and the number of prior biopsies [16]. Their evaluation of over 570 men noted a 54% sensitivity and 74% specificity of diagnosing prostate cancer on biopsy when choosing a *PCA3* cut-off score of 35. Upon further review, their AUC was 0.703 in 277 men undergoing their first prostate biopsy, compared with an AUC of 0.684 in 280 men undergoing a repeat biopsy. 

Furthermore, a recent meta-analysis involving 46 clinical trials encompassing nearly 12,300 patients utilizing the urine *PCA3* test in the diagnosis of prostate cancer was able to reproduce these findings [17]. The pooled sensitivity and specificity were 65% and 73%, respectively. When stratifying these results into initial versus repeat prostate biopsy, they found AUCs of 0.8 and 0.68, suggesting that PCA3 could potentially be more suitable for initial rather than repeat biopsy.

### 2.2. Select MDx

The SelectMDx (MDxHealth, Irvine, CA, USA) assay measures the mRNA levels of two genes, *HOXC6* and *DLX1*, that are known to be overexpressed in aggressive prostate cancer [18] (Table 1, Figure 1). The mRNA values are quantified following a post-DRE urine specimen and normalized to *KLK3* mRNA, the gene that encodes PSA. The mRNA values of *HOXC6* and *DLX1* are then combined into a single RNA value, which is then used in addition to known clinical risk factors (patient age, PSA density (PSA/prostate volume) and DRE result (normal or abnormal) to determine the percent likelihood of identifying Gleason Grade Group 2 (GGG 2) (3 + 4) or higher prostate cancer on initial prostate biopsy. The assay is a CLIA-certified lab assay and has been included in the 2020 NCCN Guidelines for Prostate Cancer Early Detection.

Two separate clinical trials demonstrated a sensitivity and NPV of ≥90% using the mRNA levels of *HOXC6* and *DLX1*. Van Neste et al. evaluated more than 900 men scheduled for either initial or repeat prostate biopsy and found an AUC of 0.76 with a sensitivity and NPV of 91 and 94%, respectively [19]. When combining these mRNA values with clinical risk factors, the AUC increased to 0.9. These results were then further validated by Haese et al. during their evaluation of 916 men undergoing initial prostate biopsy. They demonstrated an AUC of 0.85 with 93% sensitivity, 47% specificity, and 95% NPV [20]. When focusing only on the 715 men with PSA < 10 ng/mL, they obtained an AUC of 0.82 with 89% sensitivity, 53% specificity, and 95% NPV [20]. These combined results demonstrate a high sensitivity and NPV of detecting clinically significant prostate cancer prior to initial biopsy. Of note, this was a retrospective trial that was completed and there was no recorded race information reported. 

It has been shown that SelectMDx may lead to disease risk stratification of patients for multi-parametric MRI. SelectMDx was found to be more sensitive and less specific than mpMRI for detection of CaP (Select MDx vs. mpMRI; sensitivity, 86.5% vs. 51.9%/73.8%; specificity, 73.8% vs. 88.3%) and clinically significant CaP (Select MDx vs. mpMRI; sensitivity, 87.1% vs. 61.3%; specificity, 63.7% vs. 83.9%) at diagnostic biopsy in a patient cohort of 310 men [21]. A prospective study based on 599 prostate-biopsy-naïve men in the Netherlands that had PSA ≥ 3 found that 31% (183/599) of patients were diagnosed with high-grade prostate cancer. Using SelectMDx alone, 10% of those high-grade prostate cancers were not detected, while mpMRI missed 13% of high-grade prostate cancers. Select MDx (AUC 0.83) was found to perform better than PCA3 (AUC 0.65) and PSA (AUC 0.66) for predicting PIRADS 4 and 5 lesions on mpMRI. [22].

### 2.3. ExoDx Prostate (IntelliScore)

Unlike the *PCA3* and the Select MDx assays, the ExoDx prostate Intelliscore (Exosome Diagnostics Inc., Cambridge, MA, USA) is a non-DRE urine exosome-based assay that measures *PCA3* and *ERG* (V-ets erythroblastosis virus E26 oncogene homologs) RNA levels along with a control gene, *SPEDF* (Table 1, Figure 1). It then combines the molecular markers with standard of care (SOC ) variables (PSA, race, age, family history) to delineate the risk of detecting ≥ GGG 2 prostate cancer on biopsy. Currently, this assay is indicated for men ≥ 50 years of age with a PSA of 2–10 ng/mL who are scheduled for an initial prostate biopsy due to concerning DRE and/or PSA levels [23]. This test is a CLIA-certified lab assay and is included in the NCCN 2020 Guidelines for Prostate Cancer Early Detection.

A multicenter study involving 774 patients (255 training cohort, 519 validation cohort) across 22 sites within the United States demonstrated that the ExoDx assay combined with SOC variables was significantly better at predicting the presence of ≥ GGG 2 prostate cancer and negative biopsy results than either ExoDx assay or SOC variables alone (AUC combo: 0.77, gene alone: 0.74, SOC alone: 0.63) [23]. Focusing on ≥ GGG 2 prostate cancer, utilizing a predefined cut-off point of 15.6, they found a NPV of 91% and a sensitivity of 92%. Furthermore, the assay would have missed only 8% ≥ GGG 2 prostate cancer, of which only 5% had dominant pattern 4 Gleason 7 disease. Recently, they re-evaluated their data and focused on men that were presenting for repeat prostate biopsy only. Utilizing the same cut-off point of 15.6, they found an NPV and sensitivity of 92% and 82%, respectively [24]. This would have avoided 26% of unnecessary biopsies while missing only 2.1% of ≥ GGG 2 prostate cancer. 

A randomized, prospective clinical utility trial of the EPI test involving 942 patients undergoing an initial prostate biopsy after abnormal PSA found that there was a 30% higher detection of high-grade prostate cancer, and 49% fewer high-grade CaP were missed when compared to the control (SOC) in men with positive EPI scores [25]. 

One factor notable from this study was that African American males represented 23% of this total study population (22% in the EPI arm, 24% in the control arm). In the EPI arm, 91% of African American patients had elevated scores (≥15.6), which is consistent with the higher incidence of high-grade prostate cancer in this population, and it was ultimately found that the high-grade prostate cancer biopsy ratio was increased by 8% when comparing the control to EPI (35% vs. 43%).

### 2.4. PCA3-PCGEM1 Panel

The two-gene assay (*PCA3-PCGEM1*) is an exosome-based urinary assay that detects mRNA levels of *PCA3* and *PCGEM1*, normalized to a control gene, *SPDEF*. Similar to ExoDx, this also does not require a DRE prior to urine collection. The assay was developed in combination with SOC variables (PSA, age, race) to detect clinically significant cancer (GS ≥ 7) at diagnostic biopsy. *PCGEM1* was chosen in addition to *PCA3* as it has previously been demonstrated to be a prostate tissue-specific, androgen-regulated non-coding RNA that is functionally involved in the development of prostate cancer and significantly over-expressed in AA men with prostate cancer [52,53]. This of course is in contrast to the PCA3-ERG assay by ExoDx, which is an excellent assay yet may not yield as much clinical utility in AA as CA men as it was previously demonstrated that the frequency of ERG fusion is much higher in CaP in CA than AA men (50–60% vs. 25–30%, respectively). 

A single-center study involving 271 patients with either initial or prior negative prostate biopsy found that the two-gene panel combined with SOC variables was able to significantly improve the prediction of ≥GGG 2 prostate cancer on subsequent biopsy compared to SOC variables alone (AUC of 0.88 and 0.8, respectively) [54]. The study highlights the importance of developing marker panels that can be broadly applied to AA and CA men.

### 2.5. MiPS (Mi-Prostate Score)

MiPS (University of Michigan, MLabs) is a post-DRE urine-based CLIA certified lab assay that is based on multiplex analysis of *T2-ERG* fusion, *PCA3*, and serum PSA (*KLK3*) (Table 1, Figure 1). *T2-ERG* has previously been noted to represent the most common prostate-cancer-specific driver gene alteration [26]. In addition, the oncogenic activation of *ERG* through this gene fusion has been detected in 60% and 30% of prostate cancer in CA and AA men respectively [27]. A multi-institution study including a training cohort of 677 men and validation cohort of 1225, MiPS had better predictive accuracy (AUC, 0.77) for high-grade disease compared to PSA (AUC, 0.65), PSA with T2-ERG (AUC, 0.73), and PSA with PCA3 (AUC, 0.75) [28].

In another validation study based on 1525 men, MiPS (≤10) was able to avoid un-necessary biopsy in 33% of patients and missed 3% of men with GG ≥ 2. In relation to race, clinical performance and sensitivity was found to be similar for AA men (97.6% vs. 97% overall) [29. Tosoian et al. recently reported in a study based on 540 men that MiPS was significantly higher in men with GG ≥ 2 cancer than those with negative or GG1 biopsy in the overall population and when stratified by PI-RADS score [30]. In the PI-RADS 3 population (*n* = 121), MiPS showed the best clinical performance for predicting GG ≥ 2 cancer with an AUC of 0.73 compared to AUCs of 0.55 for PSA and 0.62 for PSA density. Moreover, with the test at an cut off ≤25, 39% of men in the PI-RADS 3 group could have avoided unnecessary biopsy while missing 6% of patients with GG ≥ 2 prostate cancer.

## 3. Blood-Based Diagnostic Biomarkers

### 3.1. 4Kscore Test

Due to the controversy surrounding serum PSA for prostate cancer diagnosis, the 4Kscore Test (OPKO Lab, Nashville, TN, USA) is a CLIA lab-based assay that expands upon PSA testing in an effort to improve the likelihood of detecting GG 2 or higher prostate cancer on biopsy (Table 1, Figure 1). This test combines a total of four different kallikrein proteins: total PSA, free PSA, intact PSA, and *human kallikrein 2* (*hK2*) [31] with patient age, DRE results (abnormal or normal), and results of prior prostate biopsies to provide a probability score of 0–100% of detecting clinically significant prostate cancer. The 4Kscore Test has been included in the NCCN Guidelines for Prostate Cancer Early Detection since 2015 [32].

The first prospective evaluation of the 4Kscore Test to predict clinically significant prostate cancer in the United States involved more than 1012 men and demonstrated excellent ability to predict the detection of ≥GG 2 prostate cancer on subsequent biopsy with an overall AUC of 0.82. Although no standard cut-off percent has been established in this assay, if a biopsy would have been performed for all scores greater than 9%, a total of 43% of biopsies could have been avoided and a diagnosis of ≥GG 2 prostate cancer would have been delayed in only 2.4% of patients. Among those delayed cases, 62% would have (3 + 4) disease, and only 8% would have (4 + 4) or higher disease [33]. These results were validated by a meta-analysis combining European and American studies utilizing the 4Kscore Test; it included over 11,134 patients and demonstrated an overall AUC of 0.81 [34]. Additionally, the 4Kscore Test showed a significantly higher predictive accuracy for CaP detection than PSA and DRE with an AUC of 0.68 compared to 0.58 in a cohort of 925 men with previous negative biopsy. Moreover, the AUC improved from 0.76 to 0.87 for aggressive CaP [35]. Regarding the role the 4Kscore Test across AA men, a multi-institutional validation study (N, 366 with 56% AA) based on the VA Health System by Punnen et al. showed better predictive accuracy (AUC 0.81 vs. 0.74, *p* < 0.01) and higher clinical usefulness on decision curve analysis than the base model, irrespective of race [36].

Performance of the 4Kscore Test in conjunction with mpMRI in a cohort of 266 biopsy-naïve men with a clinical suspicion of prostate cancer was found to reduce unnecessary biopsies in a study by Falagario at al. [37]. Men with low or intermediate 4Kscore Test scores (96.9% and 97.1%) showed the highest NPV for men with PI-RADS 1–2 lesions. Regarding the sequence of the procedures, the best biopsy strategy was an initial 4Kscore followed by mpMRI if the 4Kscore was ≥7.5% and a subsequent biopsy if the mpMRI was positive (PI-RADS 3–5) or the 4Kscore was ≥18%. This would lead to a missing 2.7% (2/74) of clinically significant CaP and avoiding 34.2% of biopsies.

### 3.2. Prostate Health Index (PHI)

The Prostate Health Index (PHI; Beckman Coulter, Brea, CA, USA) is used to predict the likelihood of detecting ≥ GG 2 prostate cancer at biopsy by combining the levels of free PSA, total PSA, and the [-2] form of proPSA (p2PSA) in a composite score using the following formula: ([-2]proPSA/free PSA) X √PSA (Table 1, Figure 1). The assay was FDA approved in 2012 for use in men of at least 50 years of age that have a normal DRE and a serum PSA of 4–10 ng/mL. It is included in the NCCN guidelines in 2015 to inform decision for detecting ≥ GG 2 prostate cancer at biopsy in men with an indeterminate serum PSA [32].

Catalona et al. previously described a large multicenter trial involving 892 men with PSA levels of 2–10 ng/mL and a normal DRE who underwent PHI testing prior to scheduled prostate biopsy. They demonstrated an AUC of 0.72 in discriminating ≥ GG 2 from either lower GG prostate cancer or negative biopsies. Also noted, higher PHI values were significantly associated with a higher percentage of ≥GG 2 on biopsy, ranging from 26% to 42% for PHI values < 25 and ≥55, respectively [38]. In a multicenter prospective trial, Loeb et al. reported that PHI outperformed its individual components of total, free, and [-2]proPSA for the identification of clinically significant prostate cancer (Gleason ≥ 7; AUCs phi 0.707, percent free prostate-specific antigen 0.661, [-2]proPSA 0.558, prostate specific antigen 0.551) [39]. Further, Calle et al. reported in two independent cohorts of biopsy-naïve patients that PHI detected aggressive CaP with better specificity than tPSA and %fPSA [40]. A study by Babajide et al. reported that PHI had moderate accuracy for detecting GG 2–5 CaP in Black people, but PHI cut off ≥28.0 can avoid unnecessary biopsies in Black people (sensitivity, 90.4%; avoids 17.9% un-necessary biopsies) [41]. A recent meta-analysis evaluating the diagnostic accuracy of PHI and 4Kscore noted a pooled sensitivity and specificity in detecting ≥ GG 2 prostate cancer of 0.93 and 0.34 for PHI, as well as 0.87 and 0.61 for 4Kscore, respectively. Lastly, the pooled AUC showed an accuracy of detecting high-grade prostate cancer of 0.82 and 0.81 for PHI and 4Kscore, respectively [42]. PHI (cut-off 42.7) was also found to significantly outperform both mpMRI and PHI density in the prediction of positive biopsy (PHI ≥ 61.68 and PI-RADS score ≥ 4 were able to identify csPCa (Gleason score ≥ 7 (3 + 4) both alone and added to a base model including age, PSA, fPSA-to-tPSA ratio, and prostate volume [43]. PHI outperformed MRI PIRADS score in predicting positive biopsy (AUC difference of 0.14), using a threshold score of 42.7, and that both PHI (with a cut-off threshold of 61.68) and MRI PIRADS score of 4 or greater both comparably identified the presence of clinically significant CaP at the time of prostatectomy with an AUC of 0.75. Moreover, a retrospective single-institution-based study on 2900 men showed that prebiopsy MRI led to biopsy avoidance in 31% of men. MRI usage enhanced detection of csPCa by 13% and reduced identification of GG1 disease by 3% and negative biopsies by 10% (*p* < 0.001). They also concluded that Black race was associated with reduced MRI utilization (*p* < 0.001) [44]. 

### 3.3. Circulating Tumor Cells

Most of the studies on circulating tumor cells (CTCs) are based on CaP prognosis, especially focusing on the early detection of metastatic CaP [55,56]. However, recent advancement in CTC detection may provide additional tools to detect clinically significant cancer at an early stage of CaP [57]. Under the prostate tumor early cancer test (ProTECT) study, an evaluation of 409 men for CaP screening found that the number of malignant circulating prostate cells increased significantly with age and PSA level and was associated with a cancer-positive status of biopsy. The assay achieved a sensitivity of 86.2%, specificity of 90.8%, a PPV of 78.9%, and a NPV of 94.3% for detecting CaP in men with high PSA and/or abnormal DRE [58]. Murray et al. found P504S expression to be prostate cancer specific and showed that PSA+/P504S− cells found in the peripheral blood specifically correlated with non-malignant condition in screening subjects [59]. Another study by Reid et al. suggested that the combination of cytology-based ISET^®^-CTC Test and ICC prostate marker testing had sensitivity of 97% and specificity of 99% for diagnosing CaP [60]. In a similar study by Xu et al., CTC in combination with a 12-gene panel assay achieved more than 90% accuracy for predicting CaP on subsequent biopsies [61]. In a recent prospective study by Yang et al. on 203 patients, telomerase reverse transcriptase (TERT)-based CTC detection exhibited robust diagnostic efficiency for CaP, especially for those in the PSA gray area of 4–10 ng/mL. Additionally, a combined model (CTC plus PSA) could further improve the predictive accuracy of CaP at diagnosis [62]. Although these findings are promising, more comprehensive analyses are warranted to better assess the diagnostic potential across racial groups, the additional advantage of CTCs over other biomarkers, and the standard of care variables for detecting aggressive disease at diagnosis.

## 4. Multiparametric Magnetic Resonance Imaging (mpMRI)

Prostate magnetic resonance imaging (mpMRI) is the imaging test of choice for patients suspicious of aggressive disease (Figure 1). The PI-RADS v2.1 system (five-point scale, 1–5) is used to evaluate for clinically significant disease with a score of 4 or 5 representing a high to very high likelihood of the corresponding lesion on imaging harboring clinically significant prostate cancer. The expanding field of imaging in clinical practice including mpMRI in the diagnosis and active surveillance of men with prostate cancer is yielding promising results. MRI has been shown in the PROMIS Trial to guide diagnostic pathway for men with elevated PSA. This trial showed that MRI can help in patient selection for initial biopsy and improve detection rates among those biopsied. Among 576 men who underwent upfront MRI, 27% of men were able to avoid primary biopsy and, among those biopsied on the basis of MRI-informed biopsy, up to 18% more clinically significant prostate cancers were detected, with an overall decrease by 5% of diagnosis of clinically insignificant prostate cancer [7]. Similar results were shown by the PRECISION and MRI-FIRST studies where mpMRI assessment for CaP before biopsy was found to improve the detection of clinically significant cancer [63,64]. Frost-Jan et al. also reported through a Cochrane systematic review and meta-analysis that overall MRI diagnostic pathway had the most favorable outcome in the detection of both clinically significant and insignificant prostate cancer compared with systematic biopsy [65,66]. 

Considering the clinical utility of mpMRI, it has become increasingly valuable, as reflected by NCCN, AUA, and EAU recommendations [67]. The use of mpMRI prior to confirmatory biopsy in the setting of active surveillance of known low-risk prostate cancer is now recommended by the NICE (National Institute of Health and Clinical Excellence) guidelines [68]. This is supported by the findings of Schoots et al. in a systematic review including a total of 931 active surveillance patients who underwent mpMRI prior to a confirmatory biopsy [69], mpMRI was found to be positive in 73% of patients, and 32% of patients were upgraded at confirmatory biopsy. Furthermore, they found that cancer upgrading occurred more frequently when performing systematic and targeted biopsies in comparison to either systematic or mpMRI-targeted biopsies alone (13% vs. 11% vs. 8%, respectively). The Prospective Stockholm3 Active Surveillance Trial (STHLM3AS) also showed that adding MRI-targeted biopsies to systematic biopsies increased sensitivity of GS ≥ 3 + 4 CaP compared with systematic biopsies alone. Performing biopsies in only MRI-positive instances increased the sensitivity of GS ≥ 3 + 4 CaP and reduced the number of biopsy procedures by 49.3% while missing 7.2% GS ≥ 3 + 4 CaP and 1.4% csCaP [70]. Additionally, Stanzione et al. reported that mpMRI is mainly utilized for detection and for pre-operative staging of CaP by Italian urologists; however, mpMRI availability and report standardization requires improvement [71]. MRI, therefore, has been integrated both into diagnostic pathways and into most active surveillance regimens. Changes in protocol-based MRI are often used to inform biopsies among men on active surveillance, evidenced by the independent association of increasing PIRADS score while on active surveillance with disease reclassification leading to treatment [72]. Stability of MRI lesions, or absence of lesions on serial MRIs while on active surveillance, however, has not been shown to mitigate the need for scheduled prostate biopsies, as biopsy protocols based only on imaging change miss nearly a third of disease reclassification [73]. These studies show the utility of MRI, but also the limits of imaging-only evaluation for prostate cancer diagnosis and progression.

Although MRI shows a clear benefit in the diagnosis and surveillance of prostate cancer, widespread implementation of MRI can be limited by cost, machine availability, the dependence on expert imaging interpretation, and the availability of registration software for clinical use at the time of biopsy. Standard reporting practices and implementation of AI (artificial intelligence) practices and multi-marker assessments may offer improvements and are the topic of multiple ongoing investigational efforts. Recent development on AI-based machine learning (ML) and deep learning (DL) approaches may provide a faster, automated, and precise alternative for interpretating quantitative MRI images [74]. In a multi-center study on a radiomics-powered ML approach for the detection of extraprostatic extension in CaP, the ML model had an accuracy of 83% in the training set and the support vector machine algorithm achieved accuracies of 79% and 74% in two independent institutional datasets, implying a high accuracy of the AI model along with broader utility of the model in a multicenter setting [75]. Stanzione et al. reported that a ML combined with texture analysis (TA), a quantitative postprocessing method for data extraction, appeared as a potential tool to predict histopathological EPE on biparametric MR images [76]. In a recent study, Gravina et al. showed that the ML approach using clinico-radiographic features was able to improve the specificity of mpMRI in patients with PI-RADS score 3 lesions, which could help clinicians in making diagnostic decisions including need for biopsy [77]. However, some studies have shown the limited value of ML and DL on CaP diagnosis [78]. Therefore, to better assess the role of AI in CaP diagnostic pathway, large-scale prospective and validation studies are needed.

## 5. Tissue-Based Diagnostic Biomarkers 

### Confirm MDx

Unlike the aforementioned biomarker assays, ConfirmMDx (MDxHealth, Inc, Irvine, CA, USA) is a DNA methylation assay that is prostate tissue biopsy based (Table 1, Figure 1). This test evaluates the methylation status of several genes known to be frequently found in prostate cancer: Glutathione S-Transferase Pi 1 (GSTP1), Adenomatous Polyposis Coli (APC), and Ras association domain family member 1 (RASSF1). These aforementioned markers have been demonstrated to have a “field effect”, meaning that a positive ConfirmMDx test in a cancer-negative biopsy suggests that occult cancer was missed during the prostate biopsy [45]. As such, this assay is currently included in the NCCN guidelines for management of men with elevated PSA and a prior negative prostate biopsy [46].

Wojno et al. evaluated 138 men at a total of five centers within the United States that had an elevated PSA and prior negative prostate biopsy results. Their goal was to evaluate the prevalence of repeat prostate biopsies in patients who had a negative prostate biopsy and negative Confirm MDx result. Only 6 of the 138 (4.3%) men underwent repeat prostate biopsy, all of whom had no evidence of cancer on repeat biopsy [47]. This was further evaluated in a separate study by Stewart et al. involving nearly 500 patients with a previously negative prostate biopsy, where Confirm MDx demonstrated a NPV of 90% [48]. In their multivariate analysis correcting for DRE, PSA, patient age, and histopathological characteristics, this biomarker assay was a significant independent predictor of identifying patients who would have cancer detected on repeat biopsy with an odds ratio of 3.17. 

An additional study by Partin et al. validated ConfirmMDx with a NPV of 88% (95% CI, 85–91%) prior to repeat prostate biopsy [49]. This study included 350 patients with a previously archived cancer-negative prostate biopsy sample from a total of five centers across the United States. Upon multivariate analysis including corrections for age, PSA, DRE, race, and first biopsy histopathologic characteristics, ConfirmMDx was the most significant independent predictor of patient outcome (OR 2.69, 95% CI, 1.6–4.51). In a study by Waterhouse et al. based on 211 AA men across seven urology centers across the USA, ConfirmMDx assay was found to be a useful tool for risk stratification of AA men who had an initial negative biopsy (sensitivity 74.1%; specificity 60.0%), consistent with previous studies based on CA men [50]. For detection of GS ≥ 7 CaP, sensitivity was 78% and specificity was 53%. The negative predictive values for detection of all CaP and GS ≥ 7 CaP were 78.8% and 94.2%, respectively, with an AUC of 0.76 [51].

## 6. Discussion

The emerging field of CaP biomarkers coupled with an imaging-based approach is leading towards a more comprehensive screening strategy for prostate cancer. This could help identify individuals at high risk and/or those patients harboring aggressive CaP early in the course of their disease, which can directly impact patient outcomes.

The present review defines the specific role of different assays along with imaging modalities that could help in the clinical management of a patient across AA and CA men whom the clinician deems to be suspicious of harboring undiagnosed prostate cancer, specifically focusing on those patients with an elevated serum PSA. Among those that help the clinician decide which patients need an initial prostate biopsy are PHI, 4Kscore, Select MDx, PCA3, and ConfirmMDx (Table 1, Figure 1). In contrast, others have been shown to be most useful in determining which patients needs to be re-biopsied due to the initial biopsy being negative for cancer despite continued high clinical suspicion for occult cancer: PCA3, TMPRSS2-ERG, ExoDx Intelliscore, Select MDx, and ConfirmMDx (Table 1, Figure 1). 

Although there has been significant progress in the development of CaP diagnostic markers, we lack comprehensive data on clinical trials with head-to-head comparison of these assays regarding their ability to predict clinically significant prostate cancer at initial or subsequent prostate biopsy. Scattoni et al. evaluated 211 men undergoing initial or repeat prostate biopsy with the goal of comparing the accuracy of PHI and *PCA3* in predicting detection of prostate cancer. They achieved a statistically significant difference with an overall AUC of 0.70 for PHI compared with 0.59 for *PCA3* [79]. Unfortunately, two additional studies that compared PHI and PCA3 reported mixed results. Ferro et al. demonstrated a higher AUC for PHI in a cohort of 300 men undergoing initial prostate biopsy with PSA 2–10 ng/mL, although it was not statistically significant (AUC of 0.77, PHI vs. 0.73, *PCA3*) [80]. Meanwhile, Stephan et al. reported a higher AUC in favor of *PCA3* in a patient cohort of 246 men undergoing either initial or repeat prostate biopsy, although this was also statistically insignificant (AUC of 0.74, *PCA3* vs. 0.68, PHI) [81]. In relation to the multi-marker approach including mpMRI, Pepe et al. compared mpMRI and SelectMDx for detecting clinically significant CaP among 125 men under active surveillance [56]. They found that mpMRI-targeted biopsies significantly outperformed the diagnostic accuracy of SelectMDx in the detection of clinically significantly CaP among men on active surveillance (84.9% vs. 70.3%, respectively), while other studies have shown limited benefit of mpMRI compared to SelectMDx at the initial biopsy stage [21,22].

Regarding the racial disparity in CaP, African American men have a 1.7 times higher incidence and 2.1 times higher rate of mortality from prostate cancer, yet they continue to represent less than 10% of the cohort population within prostate cancer studies [2,82]. Additionally, AA men are more likely to have advanced disease at diagnosis and also exhibit tumor upgrading from diagnostic biopsy to RP [83]. On the other hand, Asian American men are 55% less likely to be diagnosed with CaP than CA men; however, they are more likely to present with advanced CaP [84]. This contrary behavior may be attributed to lifestyle and biological factors. While the underlying reason for the increased rate of incidence, disease progression, and mortality among AA men is unknown, it is likely a combination of socioeconomic factors (economic and educational status, access to healthcare, diet) and genetics that lead to a difference in tumor biology [85,86]. In a large multiethnic population including non-Hispanic Whites, Hispanics, Asian, AA, and CA men across 98,484 incident prostate cancer cases and 8997 prostate cancer deaths from California showed that high socio-economic status (SES) was associated with higher incidence and lower mortality rates of CaP [87]. Additionally, AA men across all levels of SES carried the highest burden of CaP incidence and deaths in comparison with all other racial groups, implying that SES alone cannot entirely explain the racial differences in prostate cancer. Hansen et al., in a large prospective cohort study with 30 years of clinical follow-up, showed a higher CaP incidence and mortality in African American men and a lower incidence in Asian American men. These differences were not affected by lifestyle, diet, family history, or PSA screening [85]. A large nationwide retrospective cohort study based on 7,889,984 veterans with equal access to healthcare also showed significant disparities in the incidence of localized and metastatic CaP between AA and CA men [88]. Similar results were reported by additional studies [89,90,91]. Klebaner et al. also found a racial disparity in CaP-specific mortality in a Surveillance, Epidemiology, and End Results (SEER) program of nationally represented registry, but not in an equal access Veterans Health Administration (VHA) healthcare system [92]. Similar results were reported by Dess et al., where AA men had similar or improved CaP-specific mortality compared to CA men after adjusting for clinical variables in an equal-access healthcare system [90]. Another study based on the National Cancer Database for 526,690 patients found that overall survival disparity among men undergoing radical prostatectomy was significantly decreased, but not eliminated, for AA men, and significantly increased for Asian Americans and Pacific Islanders, in comparison with CA men after adjustments for a clinical factors and access to healthcare [93]. Although we see that SES contributes to CaP risk, the biological processes that underlie the effect of SES on CaP tumor biology remain poorly explored, partly because of a lack of standardized methods to measure SES [94,95]. Studies are underway on the impact of neighborhood level factors and chronic stress/environmental toxins on the aggressive CaP tumor biology, especially in the context of AA men [95,96].

On the other hand, considering tumor biology, many studies have demonstrated key differences between AA and CA men, but relatively few studies have included men of Asian origin [97,98]. The studies highlighted the racial disparity among men of different ancestries with respect to CaP biology, including molecular alterations in key CaP genes [99,100,101,102,103,104], DNA methylation [105,106], germline mutations [107,108,109,110], and genome-wide expression signatures [111,112,113,114]. Mahal et al. recently evaluated 2393 patients diagnosed with CaP at two large volume centers in the United States. They examined the mutational profiles of 474 genes according to race (CA, AA, or Asian) and tumor state (primary or metastatic). They found that AA men with metastatic CaP had a higher rate of mutations in the androgen receptor, DNA-repair genes, and actionable genetic mutations (mutations that are the intended target of precision-oncology drugs) [115]. These findings did not persist when evaluating the entire subset of men with primary and metastatic disease. Studies on the influence of genetics on CaP risk among immigrant men of Asian origin further support the race-associated effect on CaP tumor biology [116]. 

Thus, the impact of both socio-economic factors and genetics on tumor biology may have clinical implications on screening, diagnosis, and treatment of CaP across men of different ancestries. Considering the biological variability across men from different racial groups, molecular heterogeneity of CaP, and paucity of comparative biomarker data on minority populations, it is imperative to question the generalizability and clinical utility of the available biomarker assays across diverse racial groups. Therefore, we must place the onus on future clinical investigators to ensure that these under-represented and disparately affected AA men and less represented minority groups are appropriately represented in all future CaP biomarker studies.

## 7. Conclusions

Although it is well known that AA men possess a much higher incidence of and mortality rate from CaP, they continue to be vastly underrepresented in all CaP studies. Future clinical investigators must reverse this trend in order to achieve improved screening and treatment outcomes for this cohort. Several promising molecular biomarkers have been utilized for the detection of clinically significant CaP prior to prostate biopsy; however, no definitive conclusions could be reached regarding biomarker superiority. Further cross-validation studies and head-to-head comparisons of the potential biomarkers in well-designed and racially diverse clinical trials are needed to avoid future overdiagnosis and overtreatment of CaP as well as avoid the consequences of unnecessary prostate biopsies.

## Figures and Tables

**Figure 1 ijms-24-02185-f001:**
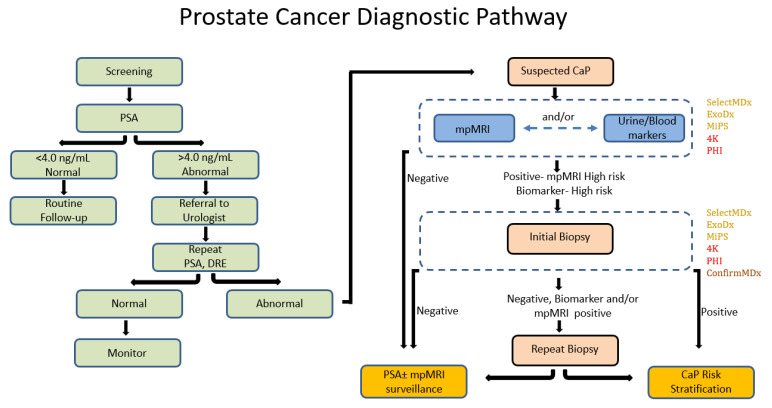
Schematic representation of traditional and proposed mpMRI-molecular-biomarker-directed prostate cancer diagnostic pathway. CaP, prostate cancer; PSA, prostate serum antigen; DRE, digital rectal examination; mpMRI, multiparametric magnetic resonance imaging. Text color of FDA/CLIA-approved molecular markers represents tissue of origin: yellow—urine derived; red—blood derived; brown—tissue derived.

**Table 1 ijms-24-02185-t001:** Prostate cancer biomarkers in the decision- making tool/diagnosis of prostate cancer categorized by molecular markers, approving body, intended use, qualifications, and clinical performance.

Biomarkers	Molecular Markers	Approval	Intended Use	Primary Qualifications	Results/Area Under the Curve (AUC)
**Urine Based Biomarkers**					
PCA3 (Progensa) [12,13,14,15,16,17]	Prostate cancer antigen 3 (*PCA3*) is a prostate specific noncoding messenger RNA (mRNA) that has been found to be over expressed in greater than 90% of all prostate tumors compared to that of benign prostatic tissue	FDA	≥50 years old Elevated PSA Prior negative biopsy	Repeat prostate biopsy	Overall: 0.68–0.87 * Initial Biopsy: 0.7–0.8 Repeat Biopsy: 0.68
SelectMDx (MDx Health) [18,19,20,21,22]	The SelectMDx (MDxHealth, Irvine, CA, USA) assay measures the mRNA levels of two genes, *HOXC6* and *DLX1*, that are known to be overexpressed in aggressive prostate cancer.	CLIA	With an elevated PSA	Initial or repeat biopsy	SelectMDx Alone: 0.76 For initial biopsy: 0.85 When combined with clinical risk factors: 0.90 For initial biopsy when PSA < 10: 0.82
ExoDx (Intelliscore) [23,24,25]	The ExoDx prostate Intelliscore (Exosome Diagnostics Inc., Cambridge, MA, USA) is a non-DRE urine exosome-based assay that measures *PCA3* and *ERG* (Vets erythroblastosis virus E26 oncogene homologs) RNA levels along with a control gene, *SPEDF*. It then combines the molecular markers with SOC (standard of care) variables (PSA, race, age, family history) to delineate the risk of detecting > GGG 2 prostate cancer on biopsy.	CLIA	≥50 years old PSA of 2–10 ng/mL Scheduled for initial or repeat biopsy.	Initial or repeat biopsy	When using combined ExoDx and SOC variables: 0.77 ExoDx alone: 0.74 SOC variables alone: 0.63 PSA alone: 0.61
MiPS [26,27,28,29,30]	MiPS (University of Michigan, MLabs) is a post-DRE urine assay which is based on multiplex analysis of *T2-ERG *fusion, *PCA3*, and serum PSA (*KLK3*).	CLIA	With an elevated PSA	Initial prostate biopsy	Overall: 0.75 For detecting > GGG2 Prostate Cancer: 0.77 (compared to using PSA alone: 0.65) For detecting > GGG2 in patients with PSA < 10: 0.88
**Serum Based Biomarkers**					
4k (OPKO Lab) [31,32,33,34,35,36,37]	Detection of 4 different kallikrein proteins: total PSA, free PSA, intact PSA, and *human kallikrein 2* (*hK2*). These values are then combined with patient age, DRE results (abnormal or normal), as well as results of prior prostate biopsies to provide a probability score of 0–100% of detecting clinically significant prostate cancer.	CLIA	With an elevated PSA	Initial or repeat prostate biopsy	Ability to detect GGG2 or greater prostate cancer: 0.81–0.82
PHI (Beckman Coulter) [38,39,40,41,42,43,44]	Analyzes the levels of free PSA, total PSA and the [−2] form of proPSA (p2PSA). It is calculated by using the following formula: ([−2] proPSA/free PSA) X √PSA.	FDA	≥50 years PSA of 4–10 ng/mL Normal DRE	Initial or repeat biopsy	Ability to detect GGG2 or higher prostate cancer: 0.72 Ability to detect high grade prostate cancer: 0.82.
Confirm MDx [45,46,47,48,49,50,51]	DNA methylation assay that is prostate tissue biopsy-based. This test evaluates the methylation status of several genes known to be frequently found in prostate cancer: *Glutathione S-Transferase Pi 1* (*GSTP1*), *Adenomatous Polyposis Coli* (*APC*), and *Ras association domain family member 1* (*RASSF1*). These markers have been demonstrated to have a “field effect”, meaning a positive ConfirmMDx test in a cancer negative biopsy suggests that occult cancer was missed during the prostate biopsy.	CLIA	With an elevated PSA Prior negative biopsy	Repeat prostate biopsy	AUC, 0.76.

* PCA3′s AUC variability due to different PCA3 cut-offs in different studies.

## Data Availability

All study data are included in the article.

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
