# Peer review of "Biomarkers of Aggressive Prostate Cancer at Diagnosis"

_ijms, 2023, doi:10.3390/ijms24032185_

Round 1

Reviewer 1 Report

Prostate cancer is very serious problem and improvement of diagnostic approach for the risk group such as african americans is topic suitable for the IJMS.

Difference in socioeconomic factors between AA and CA and their influence on the tumour biology should be discuss more detailed, optimally in the form specific subchapter, including relevant table/s.

Specifics of asian mens in the tumour biology of prostate cancer should also described and discussed.

Minor

Page 9 table did not contain any references.

Figure and table are not mentioned in the text

Reviewer 2 Report

Authors should be congratulated for their work. The topic is interesting and intriguing.  The manuscript is well written and easily readable. In order to improve the quality of the manuscript, I suggest mentioning this interesting manuscript on Prostate health index (PHI) in the PHI section (PMID: 34572950). Moreover, in order to have a complete picture of the tools to detect Prostate cancer I suggest to add a short section on mpMRI and especially on the new perspectives of Artificial intelligence in detecting the clinically significant malignancies (PMID: 30655050; PMID: 33792737: PMID: 35885471; PMID: 33348956). I also suggest to take into account the role of circulating tumor cells topic (PMID: 33958297). A major revision is required.

Round 2

Reviewer 1 Report

I have no any objections.

Reviewer 2 Report

Authors should be congratulated for their work. They improved the quality of the manuscript accordingly to all the reviewers' requests. The manuscript is suitable for publication